# AN EFFECTIVE EMBEDDING APPROACH TO SHORTEST PATH DISTANCE PREDICTION OVER LARGE-SCALE GRAPHS

## ABSTRACT

In graph data management, computing the shortest path distance between any pair of nodes is a crucial and foundational graph operation with numerous practical applications (e.g., travel/route planning, community search). Traditional algorithms for solving this problem face significant challenges in terms of time and space complexity, especially when dealing with large-scale graphs. Worse still, existing learning-based approaches often struggle with low accuracy in predicting intricate graph structures. To address these issues, this paper introduces a novel *Graph Convolutional Networks (GCN)- and Multi-View Deep Neural Networks (MVDNN)-based Distance Embedding* (GM-DE) framework, which enables the fast and accurate prediction of shortest path distances. Specifically, based on our proposed pivot and anchor set selection strategies, GM-DE enables the calculation of embeddings for each node in the graph. Then, by feeding such embeddings into our designed GCN and MVDNN models, GM-DE can be well-trained to support the mining of accurate global and local positional information for graph nodes, with the help of our constructed predictors. In this way, our GM-DE framework can achieve high accuracy in various complex scenarios, relying solely on basic node attributes as input without the need for scenario-specific data. Comprehensive experiments confirm the effectiveness and efficiency of the GM-DE approach in predicting the shortest path distances on a wide range of real-world graphs.

## 1 INTRODUCTION

Calculating the shortest path distance between two nodes is an essential task in graph data management, playing a vital role in various practical applications, including travel and route planning (Ouyang et al., 2023) and community search (Fang et al., 2016). For example, in the academic community, calculating the shortest path distance between two authors helps researchers identify communities of collaborators with close connections by analyzing connection paths, such as collaborations, conference participation, or follow-ups between researchers, which speeds up the discovery of community and user connections.

When computing the shortest path distance between two nodes within a graph, traditional algorithms have evolved into a set of classic methods over time. A prominent example is the Dijkstra algorithm (Dijkstra, 1959), a single-source shortest path algorithm. Improved algorithms, such as the label-based algorithms (Chang et al., 2012; Jin et al., 2012), determine the shortest route by marking distance information on nodes and updating the marks according to specific rules, which are more efficient in certain scenarios. However, when tackling large-scale graphs, such as those on YouTube, the shortcomings of classic algorithms become apparent. YouTube's graphs, which have billions of nodes (such as users and videos) and edges (e.g., following, watching, and commenting), pose challenges to traditional algorithms, including: i) significant time complexity, as calculating millions of paths could demand hours or even days; and ii) extensive memory use, surpassing the capacities of standard computers. Even with distributed storage, the efficiency of classic algorithms remains compromised due to the extra overhead involved in data transmission and access.

Along with the popularity of artificial intelligence techniques, various learning-based methods have been investigated to solve the shortest path problem. Specifically, learning-based methods estimate

the shortest path distance by learning potential patterns in graph data or employing distance encoding strategies (Li et al., 2020; Kolouri et al., 2021), followed by prediction models, instead of executing accurate traversal and computation. Although the results learned by these methods can significantly improve the efficiency of distance queries in the online computing stage, existing learning-based methods still suffer greatly from various limitations. Specifically, some solutions (Rizi et al., 2018; Schlötterer et al., 2019) either focus on local details or emphasize global positional information, resulting in a notable decline in prediction accuracy. Alternatively, some other methods are built on decision tree models, e.g., CatBoost (Wang et al., 2024; Jiang et al., 2021), which perform well when there are a few shortest path distance results in the graph. However, their accuracy drops significantly when faced with complex scenarios that yield numerous results. Additional, numerous existing methods (Chen et al., 2022; Huang et al., 2021; Qi et al., 2020) heavily rely on scenario-specific features. For example, numerous studies focus on road networks and incorporate geographic features such as latitude and longitude, which are heavily dependent on location, making their application to other graph structures (e.g., social networks) extremely difficult. Clearly, *there is a lack of learning-based methods that account for both accuracy and generalization ability.*

In this paper, we introduce a novel learning-based framework, named GCN-MVDNN-based Distance Embedding (GM-DE), to facilitate the estimation of shortest path distances. Based on our proposed pivot and anchor set selection methods, GM-DE enables the calculation of both local and global embeddings for each node in the graph. Meanwhile, GM-DE employs *Graph Convolutional Networks* (GCN) (Kipf & Welling, 2017) and our designed *Multi-View Deep Neural Networks* (MVDNN) to comprehensively mine the local and global positional information of all graph nodes based on our constructed predictors. Note that our GM-DE framework employs neural networks rather than decision tree models for its learning processes, thereby enhancing its ability to handle complex scenarios and maintain accuracy, even with a multitude of distance outputs. Furthermore, GM-DE significantly enhances its adaptability by using only key node attributes as input, eliminating the need for scenario-specific information. This paper makes the following four major contributions:

- We propose effective pivot and anchor set selection strategies that enable the calculation of local and global embeddings for graph nodes.

- We design MVDNN to capture full views from selected anchor sets, enhancing the capability to accurately mine the global positional information of graph nodes.

- We develop three predictors to help fuse local and global embeddings, improving GM-DE's ability to handle complex graphs without compromising accuracy.

- We conduct comprehensive experiments on five real-world graphs, showing that GM-DE outperforms state-of-the-art (SOTA) methods in both effectiveness and efficiency.

## 2 RELATED WORK

**Traditional Algorithms.** Numerous traditional algorithms and optimization approaches have been developed to solve shortest path distance problems in graphs. The Dijkstra algorithm (Dijkstra, 1959) is a classic single-source shortest path algorithm. Its core is to gradually expand from the source node through a greedy strategy, continuously determining the shortest path from each node to the source node. The time complexity can be optimized to $O(n \log n + m)$ when using a Fibonacci heap, where $n$ is the number of nodes and $m$ is the number of edges, and the space complexity is $O(n)$. The Floyd–Warshall algorithm (Floyd, 1962) is a shortest path algorithm for all pairs of vertices. Based on the dynamic programming idea, it gradually updates the distance matrix between nodes by introducing intermediate nodes to solve the shortest path distance between two nodes. Its time complexity is $O(n^3)$ and space complexity is $O(n^2)$. Label-based algorithms (Chang et al., 2012; Akiba et al., 2013) are a type of optimization algorithm whose key lies in attaching labels to each node. These labels store distance information from the node to other nodes, allowing online queries to directly obtain an approximate result with a time complexity of O(1). The node set selected for labeling directly determines the algorithm's performance, while finding the optimal node set for a graph has been proven to be an NP-hard problem. For example, in the work (Jin et al., 2012), a smaller distance error ratio will result in a denser bipartite graph, leading to a longer time required to find the optimal coverage. Meanwhile, if more distance information between nodes is stored to cover more query scenarios, the space complexity will still reach O(n²) (Jiang et al., 2014).

**Learning-based Methods.** Learning-based methods often estimate the shortest path distance by learning node embeddings. The core idea is to map the nodes into a low-dimensional vector space such that the distance or similarity between the vectors is related to the shortest path distance between the nodes. One category is based on random walk, such as Deepwalk (Perozzi et al., 2014) and node2vec (Grover & Leskovec, 2016), which generate sequences by simulating random walks and then learn node embeddings. These methods mainly capture the local positional features of nodes, but have a limited ability to estimate the shortest path distance for node pairs with long distances. One category compresses the entire distance matrix into a few elements and then reconstructs the missing elements to obtain the distance of the query node pair (Thorup & Zwick, 2005), which can be recognized as a global method. However, lossless compression of such a distance matrix is extremely challenging. A third category aims to integrate local and global information to overcome the limitations of single-perspective approaches. The method (Wang et al., 2024) resamples the probability of node occurrence in random walks and combines these embeddings with existing global embeddings to estimate the shortest path distance. However, due to the use of decision trees, it only performs well on special graphs (with a small distance range). In addition to the above three methods, many researchers have focused on specific scenarios, such as road networks. Relevant studies often leverage the characteristics of road networks to optimize algorithms, such as designing hierarchical strategies based on their hierarchical structure (Huang et al., 2021) or utilizing geographic coordinates to aid in estimation (Chen et al., 2022; Qi et al., 2020), thereby sacrificing generality in other types of graph structures.

## 3 PROBLEM DEFINITION

According to Definition 1, in our graph, the shortest path between two nodes is the path with the minimum sum of edge weights. We use the symbol $d_{i,j}$ to denote the distance of the shortest path from node $v_i$ to $v_j$ in graph $G$.

**Definition 1** *(**Graph**, $G$) An undirected graph $G = (V, E)$ consists of $V$ denoting a set of nodes $\{v_1, v_2, \ldots, v_n\}$ and $E$ denoting a set of edges $\{e_1, e_2, \ldots, e_m\}$, and each edge $(v_i, v_j)$ is assigned a weight $w_{i,j}$.*

In the following, we formally define the *Shortest-Path Distance Estimation* (SPDE) problem.

**Definition 2** *(**The Shortest-Path Distance Estimation (SPDE) Problem**, SPDE) Given a graph $G$ and two distinct nodes $v_i \in V(G)$ and $v_j \in V(G)$ in $G$, the shortest-path distance estimation (SPDE) problem returns an estimated shortest-path distance, $\hat{d}_{i,j}$, where $\hat{d}_{i,j}$ is a prediction of the shortest path distance $d_{i,j}$, such that:*

$$\hat{d}_{i,j} \approx d_{i,j}. \tag{1}$$

## 4 OUR GM-DE APPROACH

In this section, we detail our GM-DE framework, a novel approach that mines and mixes local and global positional information to address the SPDE problem over graphs. Figure 1 illustrates its core structure, which consists of three key modules: Local Embedding Generation, Global Embedding Generation, and Predictor Zoo. The local embedding module initializes the embeddings using the selected pivots and extracts the local neighborhood features of the nodes via GCN. The global embedding module captures global positional information of the nodes using MVDNN based on anchor sets. Finally, the predictor integrates the local and global embedding results through different fusion strategies to produce the final distance prediction.

### 4.1 LOCAL EMBEDDING GENERATION

**Pivot-based Embedding Initialization.** For local embedding, the initial step involves choosing a subset of nodes to serve as pivots that will initialize the embeddings for each individual node. These selected pivots help in capturing the positional embedding of each node.

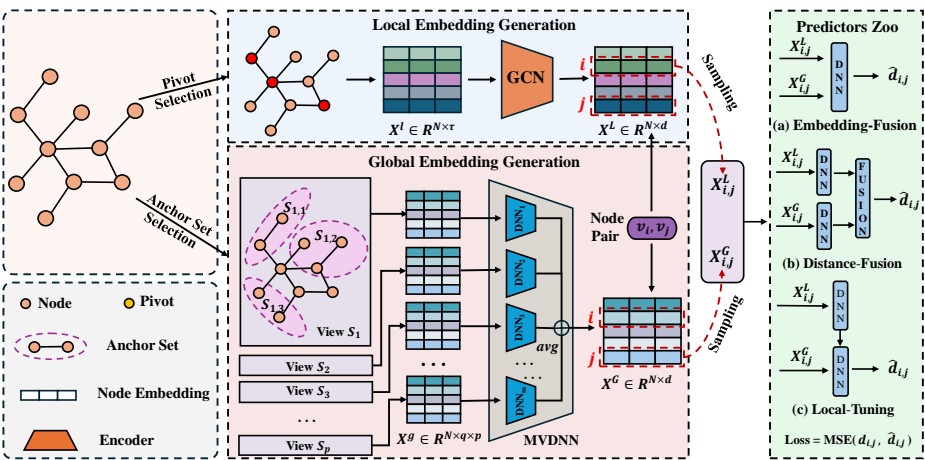

Figure 1: The framework and workflow of our GM-DE approach.

Algorithm 1 details the process of obtaining this set of pivots. First, we randomly select $\tau$ nodes as candidate pivots to form the set $P_t$ (Line 3). Then, we use Algorithm 2 to evaluate the quality of the current pivot set via $Compute\_Cost$ (Line 5). Specifically, we leverage the triangle inequality $d_{i,j} \geq |d_{i,k} - d_{k,j}|$ to compute the lower bounds of distances between sampled node pairs (Line 5 in Algorithm 2), and calculate the relative error between these lower bounds and the actual distances of these node pairs as the cost. Lower cost indicates a better pivot set. Next, we conduct multiple iterations of optimization (Lines 6-15). In each iteration, we randomly select a pivot from the set $P_t$ and then randomly choose a non-pivot node as a new candidate pivot (Lines 7-8). We replace the chosen pivot with this new candidate to form a new pivot set $P_t'$ and recompute the cost (Lines 8-9), and keep the new set if the cost is lower (Lines 11-14). This iterative process continues until the optimized pivot set $P$ is finally obtained (Lines 16-19). After obtaining the set of pivots $P$, we compute the distance from each node to each pivot to form the local embedding $X^l \in R^{N \times d}$.

---

**Algorithm 1 Pivot Selection**

---

**Input:** a graph $G$ and the number, $\tau$, of pivots
**Output:** the set, $P$, of pivots
1: $global\_cost = +\infty$, $P = \varnothing$
2: **for** $a = 1$ to $global\_iter$ **do**
3:     randomly select $\tau$ pivots, and form $P_t$
4:     uniform sampled node pairs $T$ on $G$
5:     $cost = Compute\_Cost(G, P_t)$
6:     **for** $b = 1$ to $swap\_iter$ **do**
7:         select a random pivot $piv \in P_t$
8:         randomly choose a non-pivot $new\_piv$
9:         $P_t' = P_t - \{piv\} + \{new\_piv\}$
10:        $cost\_new = Compute\_Cost(G, P_t')$
11:       **if** $cost\_new < cost$ **then**
12:          $cost = cost\_new$
13:          $P_t = P_t'$
14:       **end if**
15:     **end for**
16:     **if** $cost\_new < global\_cost$ **then**
17:         $global\_cost = cost\_new$
18:         $P = P_t$
19:     **end if**
20: **end for**
21: **return** P

---

**Encoding with GCN.** The local embedding $X^l$ is then passed through a GCN, denoted as $f_{encl}$. The GCN processes the local structure, using information propagation to encode the relationships between nodes and their neighbors. This results in a local encoded embedding $X^L \in R^{N \times d}$, which effectively captures local topological and positional information. Next, we will introduce the reasons why we chose GCN as the local encoder. Formally, given the adjacency matrix $A$ and the node feature matrix $X$, a single layer of GCN can be expressed as:

$$\mathbf{H}^{(l+1)} = \sigma\left(\tilde{\mathbf{D}}^{-\frac{1}{2}}\tilde{\mathbf{A}}\tilde{\mathbf{D}}^{-\frac{1}{2}}\mathbf{H}^{(l)}\mathbf{W}^{(l)}\right), \tag{2}$$

where $\tilde{\mathbf{A}} = \mathbf{A} + \mathbf{I}$, $\tilde{\mathbf{D}}$ denotes the diagonal degree matrix of $\tilde{\mathbf{A}}$, $\mathbf{H}^{(l)}$ represents the node representations in layer $l$ (with $\mathbf{H}^{(0)} = \mathbf{X}$ ), and $\mathbf{W}^{(l)}$ is the learnable weight matrix and $\sigma$ is a non-linear activation function. This formula introduces the essence of information propagation in GCN. That is, the updated embedding $\mathbf{H}_i^{(l+1)}$ of each node is calculated by aggregating the features of its immediate neighbors and itself through normalized operations.

In the context of our framework, this neighborhood aggregation mechanism aligns directly with the concept of local embedding. The GCN's ability to integrate adjacent node features naturally captures such localized structural contexts. Moreover, considering the static attribute of the graph in our task (i.e., the node connections remain fixed rather than changing dynamically), GCN has more advantages than graph neural networks designed for inductive tasks, such as Graph Attention Network (GAT) (Velickovic et al., 2018). Although the attention mechanism is beneficial, it introduces unnecessary computational complexity to our static graphs.

---

**Algorithm 2 Compute_Cost**

---

**Input:** a graph $G$, the set, $P$, of pivots, node pairs $T$
**Output:** the $cost$ of $P$ on $G$
1: compute the distance from each node to the $piv \in P$ as the embedding matrix $X^l$
2: $cost = 0$
3: **foreach** $(v_i, v_j, d_{i,j}) \in T$ **do**
4: $\quad d'_{i,j} = max(abs(X^l_i - X^l_j))$
5: $\quad cost+ = (d_{i,j} - d'_{i,j})/d_{i,j}$
6: **return** $cost$

---

In contrast, GCN, which uses a fixed graph structure through a normalized adjacency matrix, performs deterministic aggregation and reduces computational overhead, making it more effective in encoding stable local embeddings.

## 4.2 GLOBAL EMBEDDING GENERATION

A natural approach is to employ a graph neural network capable of aggregating global positional information as an encoder, analogous to the graph neural network used for local aggregation. However, such graph neural networks mostly generate a large amount of computational overhead and significant space costs during training. This makes it unsuitable for addressing the problem of large-scale graphs that we aim to solve. Therefore, we propose a method that employs a unique input feature selection strategy and uses MVDNN.

**Anchor-Set-based Embedding Initialization.** For global embedding, similar to the local component, we need to initialize the global embedding. To distinguish from pivots in the local component, we denote the pivot sets in the global component as anchor sets. The key difference lies in their selection methods and the information they contain. Local embedding merely serves as a coarse positional embedding of nodes, whereas global embedding, based on anchor sets, contains global positional information. Formally, let $\mathcal{S} = \{S_{1,1}, S_{1,2}, \ldots, S_{p,q}\}$ represent a collection of anchor sets, where each $S_i = \{S_{i,1}, S_{i,2}, \ldots, S_{i,q}\}$ corresponds to a view. If every node were included in the anchor sets, the embeddings would contain complete global information, but this comes with enormous computational overhead. Therefore, we rely on Bourgain's Theorem (Bourgain, 1985) below to guide the selection of anchor sets, aiming to retain as much global information as possible while controlling the overhead.

**Theorem 1** *(Bourgain's Theorem) Any finite metric space $(V, d)$ with $|V| = n$ can be embedded into a Euclidean space $\mathbb{R}^k$ (under any $\ell_p$ metric) with low distortion, where $k = O(\log^2 n)$ and the distortion is $O(\log n)$.*

Here, distortion is defined as the ratio of the embedding distances to the original distances, ensuring that the embedded space maintains the essential relationships of the original graph. Therefore, our anchor set selection strategy is designed as follows. We sample $k = p \times q$ anchor sets, where $p = \log n$ and $q = c \log n$, with $c$ being a hyperparameter. For each anchor set $S_{i,j}$ (where the view index $i \in \{1, 2, ..., \log n\}$ and the within-view set index $j \in \{1, 2, ..., c \log n\}$), each node in $V$ is included independently with probability $\frac{1}{2^i}$. This results in smaller sets (for larger $i$) providing high-certainty positional information when they contain the target node, while larger sets (for smaller $i$) have higher probabilities of containing the target node but weaker positional specificity.

For example, consider a graph with $n = 1000$ nodes. Following the strategy, we sample $p = \log_2 1000 \approx 10$ views. Since $c = 1$, the number of anchor sets per view is $q = c \log_2 1000 \approx 10$. Anchor sets $S_{1,j}$ (where $i = 1$) include each node with probability $\frac{1}{2^1}$, resulting in large sets with about 500 nodes on average. These sets contain most nodes, but provide vague positional information. In contrast, $S_{10,j}$ (where $i = 10$) include nodes with probability $\frac{1}{2^{10}}$, forming small sets with about 1 node on average. This selection strikes a balance between computational efficiency and information preservation, making MVDNN feasible for large-scale graphs.

For a node $v \in V$, its embedding in the $i$-th view is calculated based on the distances to all nodes in $S_i$. Permutation-invariant functions, such as MEAN, MIN, MAX, and SUM, can be utilized, with

non-linear transformations frequently applied before or after for enhanced expressiveness (Zaheer et al., 2017). Taking into account computational complexity, the distance from vertex $v$ to the anchor set $S_{i,j}$ is defined as the maximum distance from $v$ to any node contained within the set $S_{i,j}$:

$$d(v, S_{i,j}) = \max_{s \in S_{i,j}} d_{v,s}. \tag{3}$$

**Encoding with MVDNN.** The global embedding $X^g$ is then passed through an MVDNN (i.e., $f_{encg}$), consisting of $m$ deep neural networks. We train a deep neural network $f_i$ for $i$-th view, which maps the embedding $\{d(v, S_{i,1}), d(v, S_{i,2}), \ldots, d(v, S_{i,p})\}$ to a embedding $z_{v,i}$. To compute the ultimate embedding for node $v$, we take an average of all embeddings obtained from each view:

$$X_v^G = \frac{1}{q} \sum_{i=1}^{q} z_{v,i}. \tag{4}$$

### 4.3 PREDICTOR DESIGN

The predictor $f_{pre}$ operates by receiving local embeddings $X^L$ and global embeddings $X^G$, achieving predictions through embedding fusion and processing by a DNN or a linear layer. The DNN comprises several fully connected layers with activation functions and finally outputs $\hat{d}_{i,j}$ through the output layer. In the training phase, the Mean Squared Error (MSE) between the predicted distance $\hat{d}_{i,j}$ and the real distance $d_{i,j}$ is used as the loss function. In this paper, we construct three predictors as follows, striving to integrate local and global embeddings using different strategies.

*Embedding-Fusion (EF)* This variant concatenates local embeddings $X_i^L, X_j^L$ with global embeddings $X_i^G, X_j^G$ into a joint embedding fed into a single DNN. The DNN outputs the predicted shortest path distance $\hat{d}_{i,j}$. It enables the model to autonomously learn the association between local and global information through early embedding fusion.

*Distance-Fusion (DF)* In this variant, local embeddings $X_i^L, X_j^L$ and global embeddings $X_i^G, X_j^G$ are first input into two independent DNNs, respectively, to predict two distances $\hat{d}_{i,j}^L$ and $\hat{d}_{i,j}^G$. Then, a fusion module with a linear layer is introduced to integrate these two predicted distances. The final predicted distance is calculated as $\hat{d}_{i,j} = w_1 \cdot \hat{d}_{i,j}^L + w_2 \cdot \hat{d}_{i,j}^G$, which explicitly balances the contributions of local and global information.

*Local-Tuning (LT)* This variant first uses local embeddings $X_i^L, X_j^L$ to output a predicted local distance $\hat{d}_{i,j}^L$ through a DNN. Then, the global embeddings $X_i^G, X_j^G$ are input into another DNN with $\hat{d}_{i,j}^L$ to generate a final prediction $\hat{d}_{i,j}$, to tune the global predictions using local results.

### 4.4 IMPLEMENTATION OF GM-DE

---
**Algorithm 3 GM-DE**

---
**Input:** a graph $G$, and the number, $\theta$, of epochs
**Output:** embedding matrices $X^L$, $X^G$, and predictor $f_{pre}$
  1: initialize encoder $f_{encl}$, $f_{encg}$, and predictor $f_{pre}$
  2: select pivots, anchor sets, and training node pair set $T$
  3: compute the distance of training node pairs $d$
  4: compute the distance to pivots and anchor sets of each node as embedding matrices $X^l, X^g$
  5: **for** $epoch = 1$ $to$ $\theta$ **do**
  6:     encode $X^l$ via $f_{encl}$ for embedding matrix $X^L$
  7:     encode $X^g$ via $f_{encg}$ for embedding matrix $X^G$
  8:     feed $X^L$ and $X^G$ into $f_{pre}$ to predict the distances of training node pairs $\hat{d}$
  9:     compute $L$ with predicted distances $\hat{d}$ and actual distances $d$
 10:     update $f_{encl}$, $f_{encg}$, and predictor $f_{pre}$ minimizing $L$
 11: **end for**
 12: **return** embedding matrices $X^L, X^G$ and predictor $f_{pre}$

---

Algorithm 3 details the implementation of our GM-DE method. In Line 1, the encoder $f_{encl}$, $f_{encg}$, and predictor $f_{pre}$ are initialized. Lines 2-5 represent the computation of the distance to acquire embedding matrices $X^l$ and $X^g$ after selecting the pivots, anchor sets, and training node pairs. During the $\theta$ epochs (Lines 5-11), embedding matrices $X^l$ and $X^g$ are fed into $f_{encl}$, $f_{encg}$ (Lines 6-7), respectively, to obtain the new $X^L$ and $X^G$. Line 8 represents feeding $X^L$ and $X^G$ into $f_{pre}$ to predict the distances of the training node pairs $\hat{d}$. Finally, $f_{encl}$, $f_{encg}$, and $f_{pre}$ update to minimize the computed loss value (Lines 8-9), and return embedding matrices $X^L$, $X^G$, and predictor $f_{pre}$.

## 5 EXPERIMENTS

To evaluate our GM-DE approach, we conducted experiments using Python 3.12 on a server equipped with an Intel Core i9-10900K processor, 128 GB of memory, and an NVIDIA GeForce RTX 3080 GPU. Our experiments aim to answer the following three Research Questions (RQs):

**RQ1 (Effectiveness):** In what ways does GM-DE demonstrate improved performance in addressing the SPDE problem relative to other SOTA methods?

**RQ2 (Efficiency):** How efficient is GM-DE when it comes to generating predictions in terms of time overhead and storage costs?

**RQ3 (Benefits):** What enables GM-DE to improve the accuracy of distance predictions?

### 5.1 EXPERIMENTAL SETTINGS

**Graph Datasets.** We conducted experiments on five real-world graphs, which cover various network types, including Cora and DBLP as citation networks, Facebook and YouTube as social networks, and Dongguan as a road network. The first four datasets are all undirected and unweighted graphs, which can be regarded as graphs with edge weights set to 1. The shortest path distance between two nodes in such graphs can be calculated using the

Table 1: Statistics of datasets.

| Graph | $|V|$ | $|E|$ | $degree_{avg}$ |
|---|---|---|---|
| Cora | 2.70K | 10.7K | 3.98 |
| Facebook | 4.04K | 176K | 21.8 |
| DBLP | 317K | 1.05M | 3.31 |
| YouTube | 1.13M | 5.98M | 5.27 |
| Dongguan | 7.66K | 10.5K | 1.38 |

Breadth-First Search (BFS) algorithm. Dongguan is an undirected and weighted graph, where edge weights correspond to actual distances (in kilometers). For this graph, the shortest path distance between two nodes is computed using the Dijkstra algorithm. These graph datasets were collected from the Stanford Large Network Dataset Collection (Leskovec & Krevl, 2014) and Figshare (Karduni et al., 2016). The statistics of the graphs and the distance distributions are shown in Table 1 and Figure 2, respectively.

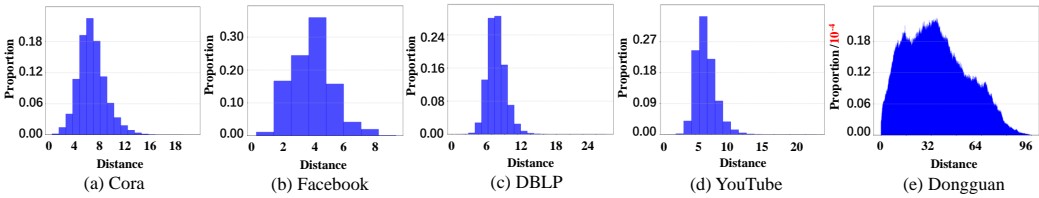

Figure 2: The shortest path distance distributions for all graph datasets.

**Baselines.** We compared our GM-DE approach with eight baseline methods, which can be classified into three categories based on the information they use. Specifically, Orion (Zhao & Zheng, 2010), Rigel (Zhao et al., 2011), and DADL (Rizi et al., 2018) belong to the first category, as they only use local information. The second category includes LS (Thorup & Zwick, 2005), ADO (Potamias et al., 2009), and P-GNN (You et al., 2019), which are designed to use global information. Path2Vec (Kutuzov et al., 2019) and BAcc (Wang et al., 2024) belong to the third category, as they combine both forms of information within their computational processes.

**Training and Testing Settings.** The parameters of each comparative method are set in accordance with their original studies to ensure optimal performance. For ours, the number of pivots is 80 for small graphs and 5 for large graphs, which corresponds to previous work, to save training cost.

Table 2: Comparison between different baselines and GM-DE, where the best performance is highlighted in bold, the runner-ups are shown in underlined, and N/A denotes "Not Applicable".

| Model | MAE | | | | | MRE | | | | |
|---|---|---|---|---|---|---|---|---|---|---|
| | Cora | Facebook | DBLP | Youtube | Dongguan | Cora | Facebook | DBLP | Youtube | Dongguan |
| Orion | 3.0542 | 1.7770 | 3.5044 | 3.5044 | 4.926 | 0.5242 | 0.6864 | 0.5165 | 0.5473 | 0.4605 |
| Rigel | 3.0426 | 1.7832 | 3.5043 | 2.8646 | 5.123 | 0.5145 | 0.6635 | 0.5164 | 0.5468 | 0.5025 |
| DADL | 1.1255 | 0.2156 | 1.2753 | **0.1568** | 4.329 | 0.2269 | 0.098 | 0.2016 | **0.0351** | 0.3990 |
| LS | 1.0599 | 0.9566 | 2.5060 | 2.0159 | 2.780 | 0.2068 | 0.3924 | 0.3939 | 0.4091 | 0.2111 |
| ADO | 2.107 | 1.1842 | 3.0691 | N/A | 2.108 | 0.4266 | 0.5080 | 0.4985 | N/A | 0.1570 |
| P-GNN | 0.5301 | 0.7502 | N/A | N/A | 2.212 | 0.1050 | 0.4065 | N/A | N/A | 0.2098 |
| Path2Vec | 3.1505 | 1.4365 | 3.9474 | N/A | 2.465 | 0.6012 | 0.5506 | 0.6097 | N/A | 0.2181 |
| BAcc | 0.8253 | **0.0149** | 0.4946 | 0.3305 | 4.035 | 0.1569 | **0.0062** | 0.0801 | 0.0801 | 0.4756 |
| GM-DE (EF) | 0.3489 | 0.1435 | **0.4682** | 0.1756 | 1.998 | 0.0785 | 0.0591 | **0.0793** | 0.0382 | 0.0554 |
| GM-DE (DF) | 0.4595 | 0.1420 | 0.5012 | 0.2012 | 2.033 | 0.0920 | 0.5880 | 0.1851 | 0.0412 | 0.0562 |
| GM-DE (LT) | **0.3296** | 0.1564 | 0.4865 | 0.1784 | **1.406** | **0.0758** | 0.0601 | 0.1768 | 0.0384 | **0.0521** |

Every GCN and DNN in our approach has two layers employing ReLU as the activation function. For training pairs, we compute the actual shortest path distances from each pivot to all node pairs using the Breadth-First Search or the Dijkstra algorithm, which yields $\tau(n - \tau)$ training pairs. In addition, Figure 2 shows the scarcity of some long-distance pairs in the training data. Therefore, we downsample the classes with few samples to balance the distribution across different distance ranges. Following the previous work (Rizi et al., 2018), for the distance categories that are extremely rare (e.g., distances 19, 20, and 21 in Cora), we directly exclude them from the training set to avoid introducing bias. For test pairs, the selection process is conducted similarly, with pivots reselected to maintain the independence of the test set. The number of node pairs is approximately 100,000.

**Evaluation Metrics.** Our evaluation metrics are divided into three main aspects: time, space, and accuracy. Let $Q$ denote a query sample set. For the time metric, since our GM-DE uses simple neural networks and efficiently samples training data, the time of training phrase at the offline stage remains within acceptable limits. Therefore, we take into account the response time of the query node pairs in the test set at the online stage. For the space metric, we evaluate the storage cost on the disk. For the accuracy metric, we assess two metrics, including *Mean Absolute Error* (MAE) and *Mean Relative Error* (MRE):

$$ MAE = \frac{1}{|Q|} \sum_{(v_i, v_j) \in Q} |\hat{d}_{i,j} - d_{i,j}|, \quad MRE = \frac{1}{|Q|} \sum_{(v_i, v_j) \in Q} \left| \frac{\hat{d}_{i,j} - d_{i,j}}{d_{i,j}} \right|. \tag{5} $$

## 5.2 COMPARISON WITH STATE-OF-THE-ARTS (RQ1)

We investigated three variants of GM-DE, with predictors of EF, DF, and LT, denoted as "GM-DE (EF)", "GM-DE (DF)", and "GM-DE (LT)", respectively. Table 2 shows that the three variants significantly outperform the baseline methods in overall performance. Compared to baseline methods that rely solely on local or global information, their MAE and MRE are significantly reduced across various datasets. For example, for the Cora dataset, the optimal variant reduces the error by more than 70% compared to traditional local methods. Compared to the hybrid method, the error is generally reduced. For large-scale datasets of DBLP and YouTube, some baseline methods cannot run effectively; however, GM-DE not only processes stably, but also achieves a significantly lower error than most baseline methods. For weighted graphs, such as the Dongguan road network, GM-DE still performs excellently, with a decrease in MAE from 2.108 to 1.406. It is worth noting that in the Facebook dataset, BAcc performs far better than other methods. The reason, as mentioned earlier, lies in the introduction of decision trees, which makes such methods effective in simple scenarios. As shown in Figure 2, Facebook has only eight distinct distance results, ranging from 1 to 8. However, in complex scenarios such as the Dongguan dataset, BAcc performs poorly.

## 5.3 INFERENCE TIME AND STORAGE COST VALIDATION (RQ2)

To evaluate the inference time of different methods, we present the time required for them to produce a prediction for the test set. Due to space constraints, only the results for the Cora dataset are presented here. Note that the results for other datasets are provided in the Appendix. As illustrated

Table 3: Comparison of the impact of local and global information on the GM-DE performance.

| Model | MAE | | | | | MRE | | | | |
|---|---|---|---|---|---|---|---|---|---|---|
| | Cora | Facebook | DBLP | Youtube | Dongguan | Cora | Facebook | DBLP | Youtube | Dongguan |
| **w/o global** | 1.0742 | 0.2885 | 1.4534 | 0.4534 | 3.407 | 0.2049 | 0.0944 | 0.2768 | 0.1583 | 0.2125 |
| **w/o local** | 0.6327 | 0.7474 | 0.6453 | 0.2547 | 2.411 | 0.1538 | 0.2845 | 0.1978 | 0.7156 | 0.1978 |
| **GM-DE (LT)** | **0.3296** | **0.1564** | **0.4865** | **0.1784** | **1.666** | **0.0758** | **0.0601** | **0.1768** | **0.0384** | **0.0521** |

in Figure 3, the inference time of the three GM-DE variants maintains a low level. Specifically, GM-DE (DF) and GM-DE (LT) take approximately 100 microseconds to complete the inference process for each query.

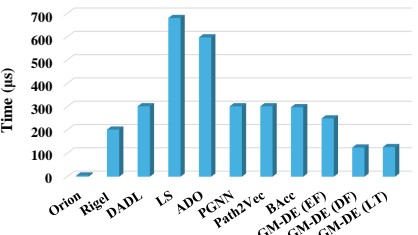 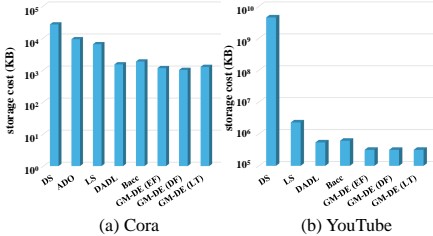

(a) Cora  (b) YouTube

Figure 3: Comparison of the inference time.  Figure 4: Comparison of the storage cost.

To evaluate the storage efficiency of different methods, we adopt Pickle serialization to quantify the storage cost of the framework. Since the storage cost of the proposed GM-DE framework is primarily determined by the embedding dimension, which is directly related to the number of pivots, we select the Cora dataset with 80 pivots and the YouTube dataset with 5 pivots as representative scenarios for comparison with other methods, and the results for other datasets are provided in the Appendix. DADL represents these learning-based methods, which have fixed feature dimensions. Traditional algorithms are typically characterized by storing the distance matrix (DS) of all node pairs. Figure 4 shows that, for the Cora dataset, the values of GM-DE (EF), GM-DE (DF), and GM-DE (LT) are relatively low, indicating that their storage requirements on this dataset are relatively small. For the YouTube dataset, the storage capacity values of these three methods are at a low level, indicating that their storage costs are relatively controllable when processing large-scale data.

## 5.4 Ablation Study (RQ3)

The ablation study evaluates the impact of local and global positional information on model performance by comparing the complete GM-DE (LT) model with variants lacking global embeddings (w/o global) and local embeddings (w/o local) across multiple datasets, as shown in Table 3. In terms of accuracy metrics, the MAE and MRE of the complete model are significantly lower than those of the two ablation variants on all datasets. This suggests that the simultaneous integration of local and global positional information can significantly enhance prediction accuracy and that the combination of the two plays a complementary role. From the perspective of the individual role of different information, comparing the two variants reveals that global positional information is more helpful to the model when predicting in most cases, aligning with expectations. However, for the Facebook dataset, w/o global performs better. The reason is that the diameter of the dataset (i.e., the maximum distance) is small. The two-layer GCN we use captures features within a two-hop range, thereby significantly improving prediction accuracy when two nodes are within four hops.

## 6 Conclusion

In this paper, we propose the GM-DE framework for the *shortest-path distance estimation* problem on large-scale graphs, addressing the critical challenges of high time/space complexity in traditional algorithms and the limited accuracy/generality of existing learning-based methods. By integrating local (via GCN) and global (via MVDNN) positional information with specialized pivot and anchor set selection, GM-DE enables fast and accurate prediction, ensuring robust generalization across diverse graphs. Experimental validation confirms that GM-DE achieves more effective and efficient predictions of the shortest path distances on various real-world graphs.

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

# A   APPENDIX

## A.1   INFERENCE TIME ANALYSIS

In RQ2 of the experiment section, we compared the inference time between all baseline methods and GM-DE methods on the dataset Cora. To further evaluate the time efficiency of GM-DE, Figure 5 supplements the results for the remaining four datasets (i.e., Facebook, DBLP, YouTube, and Dongguan) using 100,000 pairs of nodes in the query set, respectively. From this figure, we can observe that for the four datasets with different scales, the three variants of GM-DE exhibit the second-lowest inference time across all comparison methods. Note that although the baseline Orion can achieve the lowest inference time, its prediction error is about ten times larger than that of our method (see Table 2 for more details), which is not acceptable in practice. In other words, our GM-DE methods outperform all of their counterparts on the four data sets while producing minimal prediction errors, which aligns with the findings shown in Figure 3.

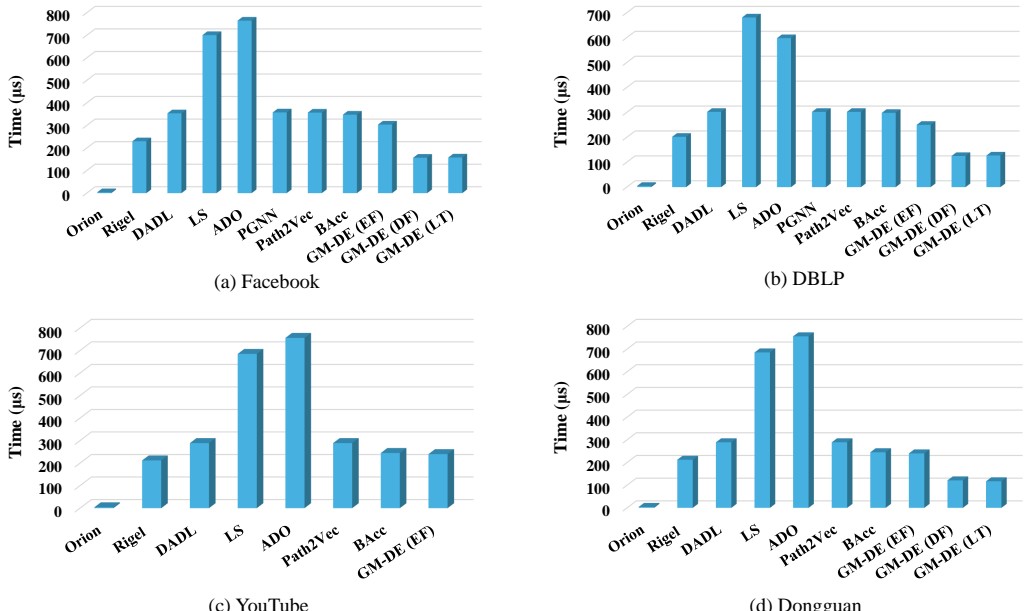

Figure 5: Comparison of the inference time.

## A.2   STORAGE COST ANALYSIS

In RQ2 of the experiment section, we compared the storage cost between all baseline methods and GM-DE methods on the datasets Cora and YouTube. To further evaluate the space efficiency of GM-DE, Figure 6 supplements the results for the remaining three datasets (i.e., Facebook, DBLP, and Dongguan). From this figure, we can observe that for the three datasets with different scales, the three variants of GM-DE require the lowest storage across all comparison methods. Please note that, since the storage cost of DADL is the same as those of Orion, Rigel, PGNN, and Path2Vec, here we only present the result of DADL. In conclusion, this experiment confirms that our strategies for selecting pivot and anchor sets can manage storage costs efficiently while maintaining prediction accuracy, thus making GM-DE methods applicable in a variety of resource-constrained settings.

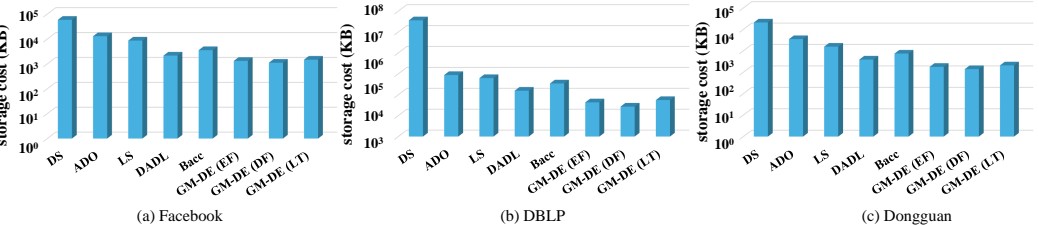

Figure 6: Comparison of the storage cost.

