# OpenReview forum: "An Effective Embedding Approach to Shortest Path Distance Prediction Over Large-Scale Graphs"
_ICLR.cc/2026/Conference — ICLR 2026 Conference Withdrawn Submission_

### Official Review · Reviewer_c5GZ · 2025-10-27

**Soundness:** 2
**Presentation:** 2
**Contribution:** 2
**Rating:** 4
**Confidence:** 3

**Summary:**

This paper presents GM-DE, a framework for shortest-path distance prediction in large-scale graphs. The method combines local and global embeddings via pivot and anchor set selection strategies and introduces three predictors to integrate these representations.

**Strengths:**

Clear Motivation and Context.

Experiments cover diverse datasets, demonstrating the method’s generality.

**Weaknesses:**

The paper is poorly organized and difficult to follow; key terms (e.g., “pivot,” “anchor set,” and “view”) are not clearly introduced or intuitively motivated.

Figure1 is not explained adequately in the text.

Although efficiency and storage are reported, the theoretical and practical scalability of the method (as $|V|$ and $|E|$ increase into the hundreds of millions/billions) is only cursorily discussed and is not validated beyond the YouTube graph (1.13M nodes).

No experiments or extrapolations are provided for truly web-scale graphs, and memory and time complexities (as well as potential GPU/CPU bottlenecks) are not analyzed in detail. Some recent works in the missing related papers category demonstrate more explicit billion-edge scalability.

**Questions:**

See Weaknesses.

---

> ### Author Response · Authors · 2025-11-28
>
> We thank the reviewer c5GZ for your insightful comments. We will address them carefully in our revised draft. Below, the "A'' refers to answer, the "W'' refers to weakness, and the "Q'' refers to question.
>
> **AW1&W2: The paper is poorly organized and difficult to follow.**
>
> We have updated our manuscript in Line 256. Figure 1 presents the
> framework and workflow of our GM-DE method, and the three modules are
> detailed in Section 4. We highlight these modules in Figure 1.
>
> **AW3&W4: The datasets are not large enough.**
>
> The selection of datasets follows \[1\]. The graph datasets are sourced
> from the Stanford Large Network Dataset Collection
> (http://snap.stanford.edu/data). In particular, YouTube and DBLP are
> regarded as large-scale graphs both in this work and in the dataset
> repository, and several baseline methods are unable to run on them.
> Consequently, experiments on these two datasets provide a meaningful
> assessment of the proposed method's scalability and robustness on
> real-world large graphs.
>
> \[1\] HaoyuWang, ChunYuan, and YuanPu. Integrating Local & Global
> Features for Estimating Shortest-path Distance in Large-scale Graphs. In
> Proceedings of the International Joint Conference on Neural Networks,
> 2024.

---

### Official Review · Reviewer_S6mk · 2025-10-28

**Soundness:** 1
**Presentation:** 2
**Contribution:** 1
**Rating:** 2
**Confidence:** 4

**Summary:**

This paper propose a graph neural network (GNN) based approach to tackle the problem of shortest path distance prediction in large-scale graphs. It propose two types pivot/anchor nodes that encode local and global features. The global features are then fused by other layers of network to predict the final distance. Experiments show improvement on predicted results compared to traditional and neural approaches, it also shows improvement on inference time.

**Strengths:**

The paper is well-motivated, with good context for the reader to understand the shortest path prediction problem.

The writing is clear and each component of the model is well-explained.

**Weaknesses:**

The proposed method provides limited contribution to both graph learning research and the shortest path distance (SPD) prediction problem.

- The method essentially adds global, local and random features as the input to a GNN, which is a paradiagm extensively explored in the GNN literature. The author did not discuss the advantage of the proposal and also did not include empirical comparison to these methods. [1, 2]

- The paper only uses GCN as the backbone GNN, whereas stronger GNN models can be just as effective without the added features. And the author did not compared to them.

- The anchor and pivot selection process is heuristic, and provides minimal insight into how selectin them bring advantage to estimiate the SPD.

- The experiment is pure transductive setting. Specifically, the test graph and train graph is exactly the same, and the model is just applied on different node pairs. 1, The GNN can simply remember a lot of features, and apply that during testing stage, without learning any transferrable knowledge. 2, If the graph is dynamic, for example, YouTube graph can change every second, does the trained model apply to the new graph? Do we need to train the entire model again? These unanswered question all limite the usability of the proposed system.

- Following last point, the transductive setting break the entire point to developing a learning system for SPD problem. One can simply do contraction hierarchy computation, and query efficiently.

- The author mentioned that the training time is within acceptable limit. This a very vague description. To show that the approach is actually useful, the author should compared their method to traditional precompute+querying approach, and show that the total time is superior rather then just inference time, especially when all experiments are conducted in a transductive fashion, and traditional approach is directly applicable.

[1] Hu, Ziniu, et al. "Pre-training graph neural networks for generic structural feature extraction." arXiv preprint arXiv:1905.13728 (2019).
[2] Abboud, Ralph, et al. "The surprising power of graph neural networks with random node initialization." arXiv preprint arXiv:2010.01179 (2020).

**Questions:**

Please see weaknesses.

---

> ### Author Response · Authors · 2025-11-28
>
> We thank the reviewer S6mk for your insightful comments. We will address them carefully in our revised draft. Below, the "A'' refers to answer, the "W'' refers to weakness, and the "Q'' refers to question.
>
>
> **AW1: Deficient baseline comparisons.**
>
> Reference \[1\] is a general pre-training framework, and \[2\] is based
> on random node embeddings. Neither is specifically designed for
> estimating the shortest-path distance. In contrast, our pivot and anchor
> sets are carefully selected for this task (local pivots are optimized
> via the triangle inequality, and global anchors follow Bourgain's
> theorem). We compared our methods with random embeddings.
>
> |   Method  | Metric |  Cora  | Facebook |  DBLP  | YouTube | Dongguan |
> |:---------:|:------:|:------:|:--------:|:------:|:-------:|:--------:|
> | GM-DE(LT) |   MAE  | 0.3296 |  0.1564  | 0.4865 |  0.1784 |   1.406  |
> |   Random  |   MAE  | 0.8788 |  0.6885  | 1.2356 |  0.4568 |   2.214  |
> | GM-DE(LT) |   MRE  | 0.0758 |  0.0601  | 0.1768 |  0.0384 |  0.0521  |
> |   Random  |   MRE  | 0.1923 |  0.3756  | 0.2589 |  0.1265 |  0.2033  |
>
> **AW2: The reasons for choosing GCN.**
>
> The main novelty of GM-DE is its integration of local and global
> embeddings through multi-view fusion. As for the selection of GCN, it is
> well-suited to the local encoding paradigm and helps lower computational
> complexity (Line 216).
>
> **AW3&W4&W5: The experiment is conducted in a pure transductive setting. The
> GNN fails to learn transferable knowledge, and the method cannot handle
> dynamic graphs.**
>
> Regarding point 1, simple embeddings or other baselines have shown that
> neural networks cannot memorize all node-pair distances. In addition,
> our training set contains distances from all nodes to only
> pivots/anchors (not all node pairs), so the training process learns
> structural patterns rather than memorizing distances. Regarding point 2,
> our problem is defined on static graphs, consistent with the majority of
> prior work. Learning on dynamic graphs is indeed a valuable future
> direction, and GM-DE can be used for fine-tuning to adapt to dynamic
> graph scenarios (e.g., efficiently updating node embeddings when edges
> are added or removed, without full retraining).
>
> **AW6: No analysis of efficiency and space complexity is provided.**
>
> The time complexity of the GM-DE framework is dominated by three stages:
> pivot/anchor set selection, embedding generation, and predictor
> training/inference. Specifically, the time complexity of pivot selection
> is $O(global\_{iter} \times swap\_{iter} \times |T|)$, where $global\_{iter}$
> and $swap\_{iter}$ are iterative optimization rounds, and $|T|$ is the
> number of sampled node pairs. Guided by Bourgain's Theorem, anchor set
> selection samples $k=O(\log^2 n)$ anchor sets ( $n$ is the number of
> nodes) with a total time complexity of $O(n\log^2 n)$. For embedding
> generation, local embedding via a two-layer GCN has a time complexity of
> $O((n + m)d)$ ( $m$ is the number of edges, $d$ is the embedding
> dimension), while global embedding through MVDNN with $q=O(\log n)$
> views has an overall time complexity of $O(n \log n \times d)$. In the
> predictor stage, the training process handles $O(\tau n)$ training pairs
> ( $\tau$ is the number of pivots) over $\theta$ epochs, resulting in a
> time complexity of $(O(\theta \times \tau n \times (d + d_{hidden}))$
> ($d_{hidden}$ is the hidden layer dimension of the predictor), and the
> inference for a single query takes $O(d)$ time with microsecond-level
> latency. In general, the offline training of GM-DE is
> $O(global\_{iter} \times swap\_{iter} \times |T| + n \log^2 n + (n + m)d + \theta \tau n d)$,
> and the complexity of the online inference is $O(d)$. Since $\tau$, $d$,
> and $\theta$ are hyperparameters independent of $n$ and $m$, GM-DE can
> efficiently adapt to large-scale graphs and support high-throughput
> queries.
>
> \[1\] Hu, Ziniu, et al. Pre-training graph neural networks for generic
> structural feature extraction. arXiv, 2019.
>
> \[2\] Abboud, Ralph, et al. The surprising power of graph neural
> networks with random node initialization. arXiv, 2020.

---

### Official Review · Reviewer_G8Qi · 2025-10-30

**Soundness:** 2
**Presentation:** 3
**Contribution:** 2
**Rating:** 4
**Confidence:** 4

**Summary:**

Traditional algorithms for calculating shortest path distances have high space and time complexity. Existing learning-based methods are inferior in accuracy due to various limitations. This paper combines graph convolutional networks (GCN) and Multi-View Deep Neural Networks (MVDNNs) based distance embeddings and proposes GM-DE framework. Experimental results demonstrate the high accuracy of the proposed GM-DE framework.

**Strengths:**

1.	This paper proposes an interesting framework to address a fundamental problem in graph data management.

2.	This paper is well written and easy to follow, especially with the well-designed illustration.

3.	The experimental results in terms of time, space, and accuracy are provided and demonstrate the advantages of GM-DE over baselines.

**Weaknesses:**

1.	The two definitions in Section 3 are trivial. The definition of shortest path estimation (Definition 2) does not involve any requirements on estimation accuracy, which seems informal.

2.	This proposed method is claimed to have advantages in efficiency over traditional methods. However, no analysis on efficiency and space complexity is provided.

3.	The proposed method is verified to be accurate and efficient in five real-world networks. However, this is not convincing for the GM-DE as a universal method for shortest distance estimation. Moreover, the tested networks are not large enough. Real-world large-scale networks contain millions or billions of nodes or edges.

4.	The traditional methods are missing in the baselines, which are important to verify the efficiency and performance advantages of the proposed method.

**Questions:**

1.	As stated, the distortion is the ratio of embedding distances to the original distances, which is O(log n). I doubt if the embedding distance follows a similar ratio scale. Otherwise, this ratio is large in terms of accurate estimation. If following the similar ratio scale (e.g., ratio > 1), how is this guaranteed in your proposed method?


2.	Please provide the time complexity analysis of the proposed framework GM-DE.

3.	Please discuss the performance guarantee of GM-DE on general real-world networks as a universal method in shortest path estimation.

---

> ### Author Response · Authors · 2025-11-28
>
> We thank the reviewer G8Qi for your insightful comments. We will address them carefully in our revised draft. Below, the "A'' refers to answer, the "W'' refers to weakness, and the "Q'' refers to question.
>
> **AW1: The two definitions in Section 3 are trivial.**
>
> The two definitions in Section 3 are introduced primarily to improve
> readability and help readers quickly understand the problem being solved
> and the notation used in the paper.
>
> **AW2&Q2: No analysis of efficiency and space complexity is provided.**
>
> The time complexity of the GM-DE framework is dominated by three stages:
> pivot/anchor set selection, embedding generation, and predictor
> training/inference. Specifically, the time complexity of pivot selection
> is $O(global\_{iter} \times swap\_{iter} \times |T|)$, where $global\_{iter}$
> and $swap\_{iter}$ are iterative optimization rounds, and $|T|$ is the
> number of sampled node pairs. Guided by Bourgain's Theorem, anchor set
> selection samples $k=O(\log^2 n)$ anchor sets ( $n$ is the number of
> nodes) with a total time complexity of $O(n\log^2 n)$. For embedding
> generation, local embedding via a two-layer GCN has a time complexity of
> $O((n + m)d)$ ( $m$ is the number of edges, $d$ is the embedding
> dimension), while global embedding through MVDNN with $q=O(\log n)$
> views has an overall time complexity of $O(n \log n \times d)$. In the
> predictor stage, the training process handles $O(\tau n)$ training pairs
> ( $\tau$ is the number of pivots) over $\theta$ epochs, resulting in a
> time complexity of $(O(\theta \times \tau n \times (d + d_{hidden}))$
> ($d_{hidden}$ is the hidden layer dimension of the predictor), and the
> inference for a single query takes $O(d)$ time with microsecond-level
> latency. In general, the offline training of GM-DE is
> $O(global\_{iter} \times swap\_{iter} \times |T| + n \log^2 n + (n + m)d + \theta \tau n d)$,
> and the complexity of the online inference is $O(d)$. Since $\tau$, $d$,
> and $\theta$ are hyperparameters independent of $n$ and $m$, GM-DE can
> efficiently adapt to large-scale graphs and support high-throughput
> queries.
>
> **AW3&Q3: The tested networks are not large enough and general**
>
> The selection of datasets follows \[1\]. The graph datasets are sourced
> from the Stanford Large Network Dataset Collection
> (http://snap.stanford.edu/data). In particular, YouTube and DBLP are
> regarded as large-scale graphs both in this work and in the dataset
> repository, and several baseline methods are unable to run on them.
> Consequently, experiments on these two datasets provide a meaningful
> assessment of the proposed method's scalability and robustness on
> real-world large graphs. The experimental datasets already cover
> representative real-world network types (social, citation, and road
> networks). The consistent superiority of GM-DE across these diverse
> graphs provides strong evidence of its effectiveness and generality as a
> shortest-path distance estimation method on real-world networks.
>
> **AW4: The traditional methods are missing in the baselines.**
>
> We recognize that traditional methods achieve 100% accuracy. However,
> their prohibitive time and space complexity (e.g., $O(n^3)$ and $O(n^3)$
> for Floyd--Warshall) make them infeasible for large-scale graphs,
> highlighting GM-DE's superiority in balancing accuracy and efficiency.
>
> **AQ1: If the embedding distance follows a similar ratio scale. And how
> is this guaranteed in the proposed method?**
>
> We do not require the learned embeddings to strictly satisfy the
> $O(\log{n})$distortion bound. It is sufficient that they preserve the
> overall trend in distance. Any scale shift can be automatically
> compensated for during neural network training.
>
> \[1\] HaoyuWang, ChunYuan, and YuanPu. Integrating Local & Global
> Features for Estimating Shortest-path Distance in Large-scale Graphs. In
> Proceedings of the International Joint Conference on Neural Networks,
> 2024.

---

### Official Review · Reviewer_7gvF · 2025-11-01

**Soundness:** 4
**Presentation:** 4
**Contribution:** 3
**Rating:** 4
**Confidence:** 3

**Summary:**

The paper proposes GM-DE, a learning-based framework for estimating shortest-path distances on large graphs by combining (i) local pivot-based embeddings refined with a GCN and (ii) global multi-view anchor-set embeddings encoded with a Multi-View DNN (MVDNN). The model trains predictors that fuse local/global embeddings in three variants (Embedding-Fusion, Distance-Fusion, Local-Tuning).

**Strengths:**

1. **Well-structured system**: The paper gives a full pipeline (pivot/anchor selection, encoders, three predictor fusion strategies) and implements them end-to-end (Alg.1–3 and training details).

2. **Empirical breadth**: Evaluation spans multiple graph types (citation, social, road) and includes both accuracy (MAE/MRE) and operational costs (inference time, storage).

3. **Ablation provided**: The “w/o local / w/o global” ablation is useful and shows the complementary value of the two components.

**Weaknesses:**

1. **Inconsistent/unclear use of Bourgain / anchor sampling**: The description of anchor sampling uses several symbols (k, p, q, c, log n, log² n) that are inconsistent/confusing. For example the paper says “we sample k = c log² n anchor sets (where i=1,...,log n, j=1,...,c log n…)” — it's unclear whether the number of views is log n or log² n, and how p and q relate. This makes it hard to reason about memory/time scaling. (Sec.4.2)
2. **Generalization and inductive behavior unclea**r: It's not clear whether GM-DE can generalize to nodes unseen during training or to changes in graph structure (edge additions). The method seems transductive (relies on precomputed distances to pivots/anchors). Explain how it handles new nodes or edges.
3. **Hyperparameters & training details missing**: Learning rates, batch sizes, number of epochs (θ), optimizer, weight decay, pivot/anchor sampling seeds, and how pivot number scales with graph size are not fully provided.

**Questions:**

1. The YouTube dataset has 1.13M nodes but only 5 pivots were used. How do pivot/anchor choices affect the results vs memory tradeoff? Also the complexity of computing distances to anchors/pivots (for large graphs) is not analyzed (offline cost can be very high if many anchors are used).

2. What are the results when you apply the method on Heterophilic Datasets?

3. GCN is used for local encoder and a DNN for each view in MVDNN. Would be helpful to compare explicitly to other positional encoders?

4 Can you compare max vs mean vs min vs quantile and learned pooling (DeepSets) for d(v,S) to show the choice’s impact?

---

> ### Author Response · Authors · 2025-11-28
>
> We thank the reviewer 7gvF for your insightful comments. We will address them carefully in our revised draft. Below, the "A'' refers to answer, the "W'' refers to weakness, and the "Q'' refers to question.
>
> **AW1: Inconsistent/unclear use of Bourgain/anchor sampling.**
>
> We have updated our paper in Line 256. We sample $k = p \times q$ anchor
> sets, where $p = \log n$ and $q = c \log n$, with $c$ being a
> hyperparameter. For each anchor set $S_{i,j}$ (where the view index
> $i \in \{1, 2, ..., \log n\}$ and the within-view set index
> $j \in \{1, 2, ..., c \log n\}$), each node in $V$ is included
> independently with probability $\frac{1}{2^i}$.
>
> **AW2: Generalization and inductive behavior unclear.**
>
> When constructing the training set, our method ensured that every node
> was represented. Since the paper does not focus on dynamic graphs, the
> graph structure remains fixed. Consequently, the method is not required
> to process previously unseen nodes or edges at inference time.
>
> **AQ1: The selection of the number of pivots/anchors.**
>
> The selection of the number of pivots/anchors is based on \[1\]. Using
> fewer pivots/anchors significantly reduces the memory cost of embeddings
> and the training and inference time. The YouTube dataset contains 1.13M
> nodes, yet only 5 pivots were used, which keeps the offline
> precomputation cost very low and fully acceptable.
>
> **AQ3: The reasons for choosing GCN.**
>
> The main novelty of GM-DE is its integration of local and global
> embeddings through multi-view fusion. As for the selection of GCN, it is
> well-suited to the local encoding paradigm and helps lower computational
> complexity (Line 216).
>
> **AQ4: The choice of the max operator.**
>
> For the global encoding step, we use the max operator as in \[2\], which
> has been shown to effectively capture long-range positional information
> while remaining computationally efficient.
>
> \[1\] HaoyuWang, ChunYuan, and YuanPu. Integrating Local & Global
> Features for Estimating Shortest-path Distance in Large-scale Graphs. In
> Proceedings of the International Joint Conference on Neural Networks,
> 2024.
>
> \[2\] Jiaxuan You, Rex Ying, and Jure Leskovec. Position-aware graph
> neural networks. In Proceedings of the International Conference on
> Machine Learning, 2019.

---

### Official Review · Reviewer_xQ15 · 2025-11-01

**Soundness:** 2
**Presentation:** 2
**Contribution:** 2
**Rating:** 2
**Confidence:** 4

**Summary:**

This work proposes GM-DE a GNN and neural network based method to estimate shortest path distances between two nodes in large-scale graphs.
The framework generates two embeddings:
* A __local embedding__ that is derived by selecting a set of pivot nodes and calculating the distance of all other nodes to them. This distance matrix $Nx\tau$ is then refined by DNN to generate the local embeddings $ X^{L} $.
* A __global embedding__ that is derived by selecting a subset of nodes to act as anchors. Then for each node it calculates the maximum distance of it to any node in the anchor-set. This distance matrix is then refined to generate the global emdding $ X^{G} $.
For each pair of nodes i,j these two embeddings are then mixed using three different strategies (each one yielding a separate model) to get the  final predicted distance.
The model is trained to minimize the MSE between the true distance $d_{i,j}$ and the predicted $\hat{d_{i,j}}$

**Strengths:**

* The proposed framework seems to outperform existing baselines and state of the art models in terms of the MSE and MAE between true and predicted distances.
* The ablation study seems to justify the choice of a local embedding $X^{L}$ and a global one $X^{G}$.

**Weaknesses:**

* While the proposed framework is quite comprehensive in the steps it takes, these steps might be over-specific and thus make it difficult to reproduce, or act as inspiration for future work. For example, the final results rely in (i) the decreasing in i size of anchor sets, (ii) the choice of max operator in eq. 3 , (iii) downsampling pairs of certain distance,etc. There is in some sense a lack of understanding or ablation studies on if and by how much those steps are necessary.
* The question of fast and approximate distance estimation mainly applies to large scale graphs it is only evaluated against two medium (by today's standards) graph sizes (YT with 5M nodes and DBLP with 300K).
* There is not enough information to evaluate the model in certain axis: (i) How much time it takes to train the model and how it scales? Either theoretically or by using synthetic graphs of increasing sizes. (ii) What is the dimension D chosen and how it affects predictions? Similar for other hyperparameters, like learning rate, exact number of samples chosen through training and how this affects performance,etc. (iii) How are the other methods tested? Do they follow the original implementation? Did the authors implement from scratch, which language, etc.

**Questions:**

See some questions from above +
* If on average on the latest category of anchor sets for i=logn, there is one node, how is the case of empty sets handled?
* In line 271 the rationale behind choosing a max aggregator is the computational complexity. How is that different from other aggregations (like min, sum, mean)? Is there any other rationale behind this choice? Like training stability, gradient considerations?
* Similarly, for (4) why is the average a good choice? If a node is the only in a set then we pretty much have all infromation about its distance to other nodes that is washed by averaging.
* For the DF measure, the true values of w1 and w2 that balance between local and global could be interesting and another addition to the ablation study.
* Line 454.. why is the inference cost directly related to the number of pivots? Should the pivots be employed just in the training phase? Then all needed is just the learnt embeddings. Similar for line 399 why are pivots selected in test set?
* Why are there NAN values for the ADO method? This is a linear method that can scale to large graphs. As a matter of fact it is tested against a graph of 49M nodes in the paper. this method is also similar to how local embeddings are obtained before the refinement of them with DNNs as happens in this work.

---

> ### Author Response · Authors · 2025-11-28
>
> We thank the reviewer xQ15 for your insightful comments. We will address them carefully in our revised draft. Below, the "A'' refers to answer, the "W'' refers to weakness, and the "Q'' refers to question.
>
> **AW1&Q2: The framework is thorough, but potentially too specific.**
>
> Regarding the anchor set sampling strategy, it strictly follows
> Bourgain's theorem (Theorem 1). This multi-scale global positioning
> approach is crucial for achieving low-distortion embeddings. For the
> global encoding step, we use the max operator as in \[1\], which has
> been shown to effectively capture long-range positional information
> while remaining computationally efficient. Regarding the downsampling of
> long-distance pairs, as illustrated in Figure 2, the distance
> distribution in the real-world graphs is highly imbalanced (e.g., in
> Cora, node pairs with distance $\geq 15$ account for only 0.3% of all
> pairs). Without such downsampling, the model would be biased toward
> short distances.
>
> **AW2: The datasets are not large enough.**
>
> The selection of datasets follows \[2\]. The graph datasets are sourced
> from the Stanford Large Network Dataset Collection
> (http://snap.stanford.edu/data). In particular, YouTube and DBLP are
> regarded as large-scale graphs both in this work and in the dataset
> repository, and several baseline methods are unable to run on them.
> Consequently, experiments on these two datasets provide a meaningful
> assessment of the proposed method's scalability and robustness on
> real-world large graphs.
>
> **AW3: There is not enough information to evaluate the model in certain
> axes.**
>
> As noted in Line 402, using a simple neural network with a sampled
> training strategy keeps offline training time entirely reasonable. Most
> hyperparameters are configured in line with previous studies, and the
> learning rate, along with other important hyperparameters, is documented
> in the paper. The baseline methods are assessed using the official
> implementations released by their original authors. For baseline methods
> without officially released source code, such as BAcc, we implement them
> in Python following the same technical environment and implementation
> standards as our GM-DE method to ensure fair comparison.
>
> **AQ1: The anchor set may be an empty set.**
>
> If an anchor set becomes empty at any point during sampling, we draw it
> again until it includes at least one node, thereby ensuring that no
> empty sets are ever used.
>
> **AQ3: The choice of the average operator.**
>
> The choice of the average as the fusion strategy in Equation (4) is
> motivated by the need to balance stability and generalization when
> integrating multi-view information. The core idea of the multi-view
> design is to use different anchor sets to capture complementary
> perspectives on the' global positions of nodes. By averaging, useful
> signals from all views are aggregated, while noise specific to
> individual views is mitigated. Even if a node appears as the only
> element in a particular anchor set, its contribution is not eliminated
> by the averaging step. The node will still co-occur with others in
> different views, and its distinctive distance characteristics will be
> maintained through multi-view fusion. In addition, using the average
> helps avoid situations in which extreme values from a single view
> dominate the final prediction.
>
> **AQ4: The true values of $w_1$ and $w_2$.**
>
> We conducted experiments to demonstrate the specific values of $w_1$and
> $w_2$, and the results are as follows:
>
> |       |  Cora  | Facebook |  DBLP  | YouTube | Dongguan |
> |:-----:|:------:|:--------:|:------:|:-------:|:--------:|
> | $w_1$ | 0.0700 |  0.4620  | 0.4865 |  0.1784 |  0.1112  |
> | $w_2$ | 0.5085 |  0.6656  | 0.6599 |  0.4356 |  0.7896  |
>
> **AQ5: Selection of pivots in the training and testing phase.**
>
> The number of pivots and anchors directly sets the embedding dimension
> D, and thus also affects the inference cost. Pivots and anchors are used
> exclusively during training to initialize embeddings and form training
> node pairs. In the online stage, only the trained embeddings and
> lightweight MLPs are required. The pivots in the test set are selected
> only to build test node pairs. We then resample them to ensure that test
> pairs do not overlap with training pairs.
>
> \[1\] Jiaxuan You, Rex Ying, and Jure Leskovec. Position-aware graph
> neural networks. In Proceedings of the International Conference on
> Machine Learning, 2019.
>
> \[2\] HaoyuWang, ChunYuan, and YuanPu. Integrating Local & Global
> Features for Estimating Shortest-path Distance in Large-scale Graphs. In
> Proceedings of the International Joint Conference on Neural Networks,
> 2024.

---

### Author Response · Authors · 2025-12-03

#

Dear Area Chairs,

Thank you for managing the review process. We sincerely thank the
reviewers for their thoughtful and constructive feedback. The rebuttal
process has significantly strengthened our paper. We have conducted a
thorough review and fully responded to the valuable opinions raised by
the experts. We summarize the key concerns raised by the reviewers and
our corresponding responses as follows.

**Clarifications:**

**Method Generalization and Inductive Ability** (W2 from 7gvF, W4 from
S6mk): The reviewers assumed that all node pairs were involved in the
training, but, in fact, only the pairs from the pivot to all the other
nodes were selected as the training node pairs. Our model is designed to
capture distance patterns between nodes and a set of anchors/pivots,
rather than memorizing distances for every possible node pair. Thus, it
can be generalized to unseen node pairs by applying the learned
structural patterns. This work is restricted to static graphs, aligning
with most previous studies, and does not consider dynamic graphs in its
problem formulation.

**Controversy over Dataset Scale** (W2 from xQ15, W3 and W4 from c5GZ): The
reviewers think that the experimental datasets are not large enough. In
fact, the dataset selection follows \[1\]. The graph datasets are
obtained from the Stanford Large Network Dataset Collection
(http://snap.stanford.edu/data). Specifically, YouTube and DBLP are
treated as large-scale graphs in this work and in the original
repository, and several baseline methods cannot be applied to them.
Therefore, the experiments conducted on these two datasets offer a
meaningful evaluation of the proposed method's scalability and
robustness on large, real-world graphs.

**Method Efficiency and Complexity** (W3 from xQ15, W2 and Q2 from G8Qi):
The complexity analysis has been specifically provided in the rebuttal
stage. In general, the offline training of GM-DE is
$O(global\_iter \times swap\_iter \times |T| + n \log^2 n + (n + m)d + \theta \tau n d)$,
and the complexity of the online inference is $O(d)$. Since $\tau$, $d$,
and $\theta$ are hyperparameters independent of $n$ and $m$, GM-DE can
efficiently adapt to large-scale graphs and support high-throughput
queries. The experimental results are presented in the paper.

**Reasons for Choosing GCN** (Q3 from 7gvF, W2 from S6mk): The reviewers
suggest employing multiple graph-based models, but the main novelty of
GM-DE lies in that it integrates local embeddings and global embeddings
through multi-view fusion. In addition, GCN is chosen because it aligns
well with the local encoding paradigm and reduces computational
complexity (Line 216).

**Overly Specific Framework** (W1 and Q3 from xQ15, Q4 from 7gvF):

-The max operator \[2\] is employed for global encoding, achieving a
trade-off between capturing long-range positional relationships and
maintaining computational efficiency.

-Long-distance pairs are downsampled because they constitute an
extremely small fraction in real-world graphs (e.g., only 0.3% of node
pairs in Cora have distance $\geq 15$), thereby helping prevent the
model from becoming biased toward learning predominantly from
short-distance pairs.

-The choice of the average as the fusion strategy in Equation (4) is
motivated by the need to balance stability and generalization when
integrating multi-view information. By averaging, useful signals from
all views are aggregated, while noise specific to individual views is
mitigated.

**Additional Experiments:**

**True Values of $w_1$ and $w_2$** (Q4 from xQ15): We conducted experiments
to demonstrate the specific values of $w_1$and $w_2$. The results are
presented in the comments, demonstrating that the global results are
dominant.

**Additional Baseline Comparisons** (Q1 from S6mk): Based on the reviewer's
suggestion, we included experiments with randomly generated embeddings,
and the results are presented in the comments.

**Contribution:**

The GM-DE framework integrates GCN and MVDNN with specialized pivot and
anchor set selection to enable fast, accurate, and generalizable
shortest path distance prediction over large-scale graphs by fusing
local and global positional information.

We believe that the revisions and additional experiments have
significantly strengthened the paper, addressing all major concerns and
providing deeper theoretical and empirical support for our claims. Thank
you again for your time and consideration.

Best Regards,\
The authors of 15609


**Reference**

\[1\] HaoyuWang, ChunYuan, and YuanPu. Integrating Local & Global
Features for Estimating Shortest-path Distance in Large-scale Graphs. In
Proceedings of the International Joint Conference on Neural Networks,
2024.

\[2\] Jiaxuan You, Rex Ying, and Jure Leskovec. Position-aware graph
neural networks. In Proceedings of the International Conference on
Machine Learning, 2019.

---

### Note · Authors · 2026-01-06

I have read and agree with the venue's withdrawal policy on behalf of myself and my co-authors.